# Projected Stein Variational Newton: A Fast and Scalable Bayesian Inference Method in High Dimensions

**Peng Chen, Keyi Wu, Joshua Chen, Thomas O'Leary-Roseberry, Omar Ghattas**
Oden Institute for Computational Engineering and Sciences
The University of Texas at Austin
Austin, TX 78712.
{peng, keyi, joshua, tom, omar}@oden.utexas.edu

## Abstract

We propose a projected Stein variational Newton (pSVN) method for high-dimensional Bayesian inference. To address the curse of dimensionality, we exploit the intrinsic low-dimensional geometric structure of the posterior distribution in the high-dimensional parameter space via its Hessian (of the log posterior) operator and perform a parallel update of the parameter samples projected into a low-dimensional subspace by an SVN method. The subspace is adaptively constructed using the eigenvectors of the averaged Hessian at the current samples. We demonstrate fast convergence of the proposed method, complexity independent of the parameter and sample dimensions, and parallel scalability.

## 1 Introduction

Bayesian inference provides an optimal probability formulation for learning complex models from observational or experimental data under uncertainty by updating the model parameters from their prior distribution to a posterior distribution [30]. In Bayesian inference we typically face the task of drawing samples from the posterior probability distribution to compute various statistics of some given quantities of interest. However, this is often prohibitive when the posterior distribution is high-dimensional; many conventional methods for Bayesian inference suffer from the curse of dimensionality, i.e., computational complexity grows exponentially or convergence deteriorates with increasing parameter dimension.

To address this curse of dimensionality, several efficient and dimension-independent methods have been developed that exploit the intrinsic properties of the posterior distribution, such as its smoothness, sparsity, and intrinsic low-dimensionality. Markov chain Monte Carlo (MCMC) methods exploiting geometry of the log-likelihood function have been developed [16, 21, 24, 12, 3], providing more effective sampling than black-box MCMC. For example, the DILI MCMC method [12] uses the low rank structure of the Hessian of the negative log likelihood in conjunction with operator-weighted proposals that are well-defined on function space to yield a sampler whose performance is dimension-independent and effective at capturing information provided by the data. However, despite these enhancements, MCMC methods remain prohibitive for problems with expensive-to-evaluate likelihoods (i.e., involving complex models) and in high parameter dimensions. Deterministic sparse quadratures were developed in [28, 26, 8] and shown to converge rapidly with dimension-independent rates for smooth and sparse problems. However, the fast convergence is lost when the posterior has significant local variations, despite enhancements with Hessian-based transformation [27, 9].

Variational inference methods reformulate the sampling problem as an optimization problem that approximates the posterior by minimizing its Kullback–Leibler divergence with a transformed prior

[22, 20, 4], which can be potentially much faster than MCMC. In particular, Stein variational methods, which seek a composition of a sequence of simple transport maps represented by kernel functions using gradient descent (SVGD) [20, 11, 19] and especially Newton (SVN) [14] optimization methods, are shown to achieve fast convergence in relatively low dimensions. However, these variational optimization methods can again become deteriorated in convergence and accuracy in high dimensions. The curse of dimensionality can be partially addressed by a localized SVGD on Markov blankets, which relies on a conditional independence structure of the target distribution [32, 31].

**Contributions**: In this work, we develop a projected Stein variational Newton method (pSVN) to tackle the challenge of high-dimensional Bayesian inference by exploiting the intrinsic low-dimensional geometric structure of the posterior distribution (where it departs from the prior), as characterized by the dominant spectrum of the prior-preconditioned Hessian of the negative log likelihood. This low-rank structure, or fast decay of eigenvalues of the preconditioned Hessian, has been proven for some inference problems and commonly observed in many others with complex models [5, 6, 29, 18, 12, 9, 10, 2, 7]. By projecting the parameters into this data-informed low-dimensional subspace and applying the SVN in this subspace, we can effectively mitigate the curse of dimensionality. We demonstrate fast convergence of pSVN that is independent of the number of parameters and samples. In particular, in two (both linear and nonlinear) experiments we show that the intrinsic dimension is a few (6) and a few tens (40) with the nominal dimension over 1K and 16K, respectively. We present a scalable parallel implementation of pSVN that yields rapid convergence, minimal communication, and low memory footprint, thanks to this low-dimensional projection.

Below, we present background on Bayesian inference and Stein variational methods in Section 2, develop the projected Stein variational Newton method in Section 3, and provide numerical experiments in Section 4.

## 2 Background

### 2.1 Bayesian inference

We consider a random parameter $x \in \mathbb{R}^d$, $d \in \mathbb{N}$, with a prior probability density function $p_0 : \mathbb{R}^d \to \mathbb{R}$, and noisy observational data $y$ of a parameter-to-observable map $f : \mathbb{R}^d \to \mathbb{R}^s$, $s \in \mathbb{N}$, i.e.,

$$y = f(x) + \xi, \tag{1}$$

where $\xi \in \mathbb{R}^s$ represents observation noise with probability density function $p_\xi : \mathbb{R}^s \to \mathbb{R}$. The posterior density $p(\cdot|y) : \mathbb{R}^d \to \mathbb{R}$ of $x$ conditioned on the data $y$ is given by Bayes' rule

$$p(x|y) = \frac{1}{Z} p_y(x), \quad \text{where } p_y(x) := p_\xi(y - f(x)) \, p_0(x), \tag{2}$$

and the normalization constant $Z$, typically $Z \neq 1$ if $p_\xi$ or $p_0$ is known up to a constant, is given by

$$Z := \mathbb{E}_{p_0}[p_\xi(y - f(x))] = \int_{\mathbb{R}^d} p_\xi(y - f(x)) p_0(x) dx. \tag{3}$$

In practice, $Z$ is computationally intractable, especially for large $d$.

### 2.2 Stein variational methods

While sampling from the prior is tractable, sampling from the posterior is a great challenge. One method to sample from the posterior is to find a transport map $T : \mathbb{R}^d \to \mathbb{R}^d$ in a certain function class $\mathcal{T}$ that pushes forward the prior to the posterior by minimizing a Kullback–Leibler (KL) divergence

$$\min_{T \in \mathcal{T}} \mathcal{D}_{\mathrm{KL}}(T_* p_0 | p_y). \tag{4}$$

Stein variational methods [20, 14] simplify the minimization of (4) for one possibly very complex and nonlinear transport map $T$ to a sequence of simpler transport maps that are perturbations of the identity, i.e., $T = T_L \circ T_{L-1} \circ \cdots \circ T_2 \circ T_1$, $L \in \mathbb{N}$, where

$$T_l(x) = I(x) + \varepsilon Q_l(x), \quad l = 1, \ldots, L, \tag{5}$$

with $I(x) = x$, step size $\varepsilon$, and perturbation map $Q_l : \mathbb{R}^d \to \mathbb{R}^d$. Let $p_l$ denote the pushforward density $p_l := (T_l \circ \cdots \circ T_1)_* p_0$. For $l = 1, 2, \ldots$, we define a cost functional $\mathcal{J}_l(Q)$ as

$$\mathcal{J}_l(Q) := \mathcal{D}_{\mathrm{KL}}((I + Q)_* p_{l-1} | p_y). \tag{6}$$

Then at step $l$, Stein variational methods lead to

$$Q_l = -\mathcal{H}_l^{-1} \nabla \mathcal{J}_l(0), \tag{7}$$

where $\nabla \mathcal{J}_l(0) : \mathbb{R}^d \to \mathbb{R}^d$ is the Fréchet derivative of $\mathcal{J}_l(Q)$ evaluated at $Q = 0$, and $\mathcal{H}_l$ is a preconditioner. For the SVGD method [20], $\mathcal{H}_l = I$, while for the SVN method [14], $\mathcal{H}_l \approx \nabla^2 \mathcal{J}_l(0)$, an approximation of the Hessian of the cost functional $\nabla^2 \mathcal{J}_l(0)$.

Given basis functions $k_n : \mathbb{R}^d \to \mathbb{R}$, $n = 1, \ldots, N$, an ansatz representation of $Q_l$ is defined as

$$Q_l(x) = \sum_{n=1}^{N} c_n k_n(x), \tag{8}$$

where $c_n \in \mathbb{R}^d$, $n = 1, \ldots, N$, are unknown coefficient vectors. It is shown in [14] that the coefficient vector $\boldsymbol{c} = (c_1^\top, \ldots, c_N^\top)^\top \in \mathbb{R}^{Nd}$ is a solution of the linear system

$$\mathbb{H}\boldsymbol{c} = -\boldsymbol{g}, \tag{9}$$

where $\boldsymbol{g} = (g_1^\top, \ldots, g_N^\top)^\top \in \mathbb{R}^{Nd}$ is the gradient vector given by

$$g_m := \mathbb{E}_{p_{l-1}}[-\nabla_x \log(p_y) k_m - \nabla_x k_m], \quad m = 1, \ldots, N, \tag{10}$$

and $\mathbb{H} \in \mathbb{R}^{Nd \times Nd}$ is the Hessian matrix, specified as the identity for SVGD [20], which leads to $c_n = -g_n$, $n = 1, \ldots, N$, while for SVN it is given with $mn$-block $\mathbb{H}_{mn} \in \mathbb{R}^{d \times d}$ by [14]

$$\mathbb{H}_{mn} := \mathbb{E}_{p_{l-1}}[-\nabla_x^2 \log(p_y) k_n k_m + \nabla_x k_n (\nabla_x k_m)^\top], \quad m, n = 1, \ldots, N. \tag{11}$$

At each step $l = 1, 2, \ldots$, the expectation $\mathbb{E}_{p_{l-1}}[\cdot]$ in (10) and (11) are approximated by the sample average approximation with samples $x_1^{l-1}, \ldots, x_N^{l-1}$, which are drawn from the prior at $l = 1$ and pushed forward by (5) once the coefficients $c_1, \ldots, c_N$ are obtained. We remark that in the original SVGD method [20], the samples are moved with the simplified perturbation $Q_l(x_m) = c_m$.

In both [20] and [14], the basis functions $k_n(x)$ are specified by a suitable kernel function $k_n(x) = k(x, x')$ at $x' = x_n$, $n = 1, \ldots, N$, e.g., a Gaussian kernel given by

$$k(x, x') = \exp\left(-\frac{1}{2}(x - x')^\top M (x - x')\right), \tag{12}$$

where $M$ is a metric that measures the distance between $x$ and $x' \in \mathbb{R}^d$. In [20], it is specified as rescaled identity matrix $\alpha I$ for $\alpha > 0$ depending on the samples, while in [14], $M$ is given by $M = \mathbb{E}_{p_{l-1}}[-\nabla_x^2 \log(p_y)]/d$ to account for the geometry of the posterior by averaged Hessian information. This was shown to accelerate convergence for both SVGD and SVN compared to $\alpha I$. We remark that for high-dimensional complex models where a direct computation of the Hessian $\nabla_x^2 \log(p_y)$ is not tractable, its low-rank decomposition by randomized algorithms can be applied.

## 3  Projected Stein variational Newton

### 3.1  Dimension reduction by projection

Stein variational methods suffer from the curse of dimensionality, i.e., the sample estimate (e.g., for variance) deteriorates considerably in high dimensions because the global kernel function (12) cannot represent the transport map well, as shown in [32, 31] for SVGD. This challenge can be alleviated in moderate dimensions by a suitable choice of the metric in (12) as demonstrated in [14]. However it is still present when the dimension becomes high. An effective method to tackle this difficulty, which relies on conditional independence of the posterior density, uses local kernel functions defined over a Markov blanket with much lower dimension, thus achieving effective dimension reduction [32, 31].

In many applications, even when the nominal dimension of the parameter is very high, the intrinsic parameter dimension informed by the data is typically low, i.e., the posterior density is effectively different from the prior density only in a low-dimensional subspace [5, 6, 29, 18, 12, 9, 10, 2]. This is because: (i) the prior $p_0$ may have correlation in different dimensions, (ii) the parameter-to-observable map $f$ may be smoothing/regularizing, (iii) the data $y$ may not be very informative, or a combined

effect. Let $\Psi = (\psi_1, \ldots, \psi_r) \in \mathbb{R}^{d \times r}$ denote the basis of a subspace of dimension $r \ll d$ in $\mathbb{R}^d$. Then we can project the parameter $x$ with mean $\bar{x}$ into this subspace as

$$x^r = \bar{x} + P_r(x - \bar{x}) = \bar{x} + \sum_{i=1}^{r} \psi_i(\psi_i, (x - \bar{x}))_H = \bar{x} + \sum_{i=1}^{r} \psi_i w_i = \bar{x} + \Psi w, \quad (13)$$

where $w = (w_1, \ldots, w_r) \in \mathbb{R}^r$ is a vector of coefficients $w_i = (\psi_i, x - \bar{x})_H$ of the projection of $x - \bar{x}$ to $\psi_i$ in a suitable norm $H$, e.g., $(\psi_i, x - \bar{x})_H = \psi_i^T \Gamma_0^{-1}(x - \bar{x})$ where $\Gamma_0$ is the prior covariance of $x$ and $\psi_i^T \Gamma_0^{-1} \psi_j = \delta_{ij}$. We define the projected posterior as

$$p^r(x|y) = \frac{1}{Z^r} p_y^r(x), \quad \text{where } p_y^r(x) = p_\xi(y - f(x^r))p_0(x) \text{ and } Z^r = \mathbb{E}_{p_0}[p_\xi(y - f(x^r))]. \quad (14)$$

Then we can establish convergence under the following assumption. We define $||\cdot||_X$ as a suitable norm, e.g., $||x||_X^2 = x^T X x$ with $X = I$, the identity matrix or a mass matrix discretized from identity operator in finite dimension approximation space in our numerical experiments.

**Assumption 1.** *For Gaussian noise $\xi \in \mathcal{N}(0, \Gamma)$ with s.p.d. covariance $\Gamma \in \mathbb{R}^{s \times s}$. Let $||v||_\Gamma := (v^T \Gamma^{-1} v)^{1/2}$ for any $v \in \mathbb{R}^s$. Assume there exists a constant $C_f > 0$ such that for any $x^r$ in (13)*

$$\mathbb{E}_{p_0}[||f(x^r)||_\Gamma] \leq C_f \quad \text{and} \quad \mathbb{E}_{p_0}[||f(x)||_\Gamma] \leq C_f. \quad (15)$$

*For every $b > 0$, assume there is $C_b > 0$ such that for all $x_1, x_2$ with $\max\{||x_1||_X, ||x_2||_X\} < b$,*

$$||f(x_1) - f(x_2)||_\Gamma \leq C_b ||x_1 - x_2||_X. \quad (16)$$

We state the convergence result for the projected posterior density in the following theorem, whose proof is presented in Appendix A.

**Theorem 1.** *Under Assumption 1, there exists a constant $C$ independent of $r$ such that*

$$\mathcal{D}_{KL}(p(x|y) \,|\, p^r(x|y)) \leq C ||x - x^r||_X. \quad (17)$$

**Remark 1.** *Theorem 1 indicates that the projected posterior converges to the full one as along as the projected parameter converges in $X$-norm, and that the convergence of the former is bounded by the latter. In practical applications, the former may converge faster than the latter because it only depends on the data-informed subspace while the latter is measured in data-independent $X$-norm.*

## 3.2 Projected Stein variational Newton

Let $p_0^r$ denote the prior densities for $x^r$ in (13). Let $x^\perp = x - x^r$. Then the prior is decomposed as

$$p_0(x) = p_0^r(x^r)p_0^\perp(x^\perp|x^r), \quad (18)$$

where $p_0^\perp(x^\perp|x^r)$ is a conditional density, which becomes $p_0^\perp(x^\perp)$ if $p_0$ is a Gaussian density. Then the projected posterior density $p_y^r(x)$ in (14) becomes

$$p_y^r(x) = p_\xi(y - f(x^r))p_0^r(x^r)p_0^\perp(x^\perp|x^r), \quad (19)$$

so that sampling from $p_y^r(x)$ can be realized by sampling from $p_y^r(x^r) = p_\xi(y - f(x^r))p_0^r(x^r)$ for $x^r$ and from $p_0^\perp(x^\perp|x^r)$ for $x^\perp$ conditioned on $x^r$ (or from $p_0^\perp(x^\perp)$ if $p_0$ is Gaussian). To sample from the posterior, we can sample $x$ from the prior, decompose it as $x = x^r + x^\perp$, freeze $x^\perp$, push $x^r$ to $x_y^r$ as a sample from $p_y^r(x^r)$, and construct the posterior sample as $x_y = x_y^r + x^\perp$.

To sample from $p_y^r(x^r)$ in the projection subspace, we seek a transport map $T$ that pushes forward $p_0^r(x^r)$ to $p_y^r(x^r)$ by minimizing the KL divergence between them. Since the randomness of $x^r = \bar{x} + \Psi w$ is fully represented by $w$ given the projection basis $\Psi$, we just need to find a transport map that pushes forward $\pi_0(w) = p_0^r(x^r)$ to $\pi_y(w) = p_y^r(x^r)$ in the (coefficient) parameter space $\mathbb{R}^r$, where $r \ll d$. Similarly in the full space, we look for a composition of a sequence of maps $T = T_L \circ T_{L-1} \circ \cdots \circ T_2 \circ T_1, L \in \mathbb{N}$, with

$$T_l(w) = I(w) + \varepsilon Q_l(w), \quad l = 1, \ldots, L, \quad (20)$$

where the perturbation map $Q_l$ is represented by the basis functions $k_n : \mathbb{R}^r \to \mathbb{R}, n = 1, \ldots, N$, as

$$Q_l(w) = \sum_{n=1}^{N} c_n k_n(w), \quad (21)$$

Then the coefficient vector $\boldsymbol{c} = ((c_1)^\top, \ldots, (c_N)^\top)^\top \in \mathbb{R}^{Nr}$ is the solution of the linear system

$$\mathbb{H}\boldsymbol{c} = -\boldsymbol{g}. \tag{22}$$

Here the $m$-th component of the gradient $\boldsymbol{g}$ is defined as

$$g_m := \mathbb{E}_{\pi_{l-1}}[-\nabla_w \log(\pi_y)k_m - \nabla_w k_m], \tag{23}$$

and the $mn$-th component of the Hessian $\mathbb{H}$ for pSVN is defined as

$$\mathbb{H}_{mn} := \mathbb{E}_{\pi_{l-1}}[-\nabla_w^2 \log(\pi_y)k_n k_m + \nabla_w k_n(\nabla_w k_m)^\top]. \tag{24}$$

The expectations in (23) and (24) are evaluated by sample average approximation at samples $w_1^{l-1}, \ldots, w_N^{l-1}$, which are drawn from $\pi_0$ for $l = 1$ and pushed forward by (20) as $w_n^l = T(w_n^{l-1})$, $n = 1, \ldots, N$. By the definition of the projection (13), we have

$$\nabla_w \log(\pi_y(w)) = \Psi^\top \nabla_x \log(p_y^r(x^r)), \text{ and } \nabla_w^2 \log(\pi_y(w)) = \Psi^\top \nabla_x^2 \log(p_y^r(x^r))\Psi. \tag{25}$$

For the basis functions $k_n$, $n = 1, \ldots, N$, we use a Gaussian kernel $k_n(w) = k(w, w_n)$ defined as in (12), with the metric $M$ given by an averaged Hessian at the current samples $w_1^{l-1}, \ldots, w_N^{l-1}$, i.e.,

$$M = -\frac{1}{r}\mathbb{E}_{\pi_{l-1}}[\nabla_w^2 \log(\pi_y)] \approx -\frac{1}{rN}\sum_{n=1}^{N} \nabla_w^2 \log(\pi_y(w_n^{l-1})). \tag{26}$$

We remark that the projected system (22) is of size $Nr \times Nr$, which is a considerable reduction from the full system (9) of size $Nd \times Nd$, since $r \ll d$. To further reduce the size of the coupled system (22), we use a classical "mass-lumping" technique to decouple it as $N$ systems of size $r \times r$

$$\mathbb{H}_m c_m = -g_m, \ m = 1, \ldots, N, \tag{27}$$

where $g_m$ is given as in (25), and $\mathbb{H}_m$ is given by the lumped Hessian

$$\mathbb{H}_m := \sum_{n=1}^{N} \mathbb{H}_{mn}, \ m = 1, \ldots, N, \tag{28}$$

with $\mathbb{H}_{mn}$ defined in (24). We refer to [14] for this technique and a diagonalization $\mathbb{H}_m = \mathbb{H}_{mm}$. Moreover, to find a good step size $\varepsilon$ in (20), we adopt a classical line search [23], see Appendix B.

### 3.3 Hessian-based subspace

To construct a data-informed subspace of the parameter space, we exploit the geometry of the posterior density characterized by its Hessian. More specifically, we seek the basis functions $\psi_i$, $i = 1, \ldots, r$, as the eigenvectors corresponding to the $r$ largest eigenvalues of the generalized eigenvalue problem

$$\mathbb{E}[\nabla_x^2 \eta_y(x)]\psi_i = \lambda_i \Gamma_0^{-1}\psi_i, \quad i = 1, \ldots, r, \tag{29}$$

where $\Gamma_0$ is the covariance of $x$ under the prior distribution (not necessarily Gaussian), $\psi_i^T \Gamma_0^{-1}\psi_j = \delta_{ij}$, $i, j = 1, \ldots, r$, $\mathbb{E}[\nabla_x^2 \eta_y(x)]$, with $\eta_y(x) := -\log(p_\xi(y - f(x)))$, is the averaged Hessian of the negative log-likelihood function w.r.t. a certain distribution, e.g., the prior, posterior, or Gaussian approximate distribution [13]. Here we propose to evaluate $\mathbb{E}[\nabla_x^2 \eta_y(x)]$ by an adaptive sample average approximation at the samples pushed from the prior to the posterior, and adaptively construct the eigenvectors $\Psi$, as presented in next section. For linear Bayesian inference problems, with $f(x) = Ax$ for $A \in \mathbb{R}^{s \times d}$, a Gaussian prior distribution $x \sim \mathcal{N}(\bar{x}, \Gamma_0)$ and a Gaussian noise $\xi \sim \mathcal{N}(0, \Gamma_\xi)$ lead to a Gaussian posterior distribution given by $\mathcal{N}(x_{\text{MAP}}, \Gamma_{\text{post}})$, where [30]

$$\Gamma_{\text{post}}^{-1} = \nabla_x^2 \eta_y + \Gamma_0^{-1}, \ x_{\text{MAP}} = \bar{x} - \Gamma_{\text{post}}A^T \Gamma_\xi^{-1}(y - A\bar{x}). \tag{30}$$

Therefore, the eigenvalue $\lambda_i$ of $(\nabla_x^2 \eta_y, \Gamma_0^{-1})$, with $\nabla_x^2 \eta_y = A^T \Gamma_\xi^{-1}A$, measures the relative variation between the data-dependent log-likelihood and the prior in direction $\psi_i$. For $\lambda_i \ll 1$, the data provides negligible information in direction $\psi_i$, so the difference between the posterior and the prior in $\psi_i$ is negligible. In fact, it is shown in [29] that the subspace constructed by (29) is optimal for linear $f$. Let $(\lambda_i, \psi_i)_{1 \leq i \leq r}$ denote the $r$ largest eigenpairs such that $|\lambda_1| \geq |\lambda_2| \geq \cdots \geq |\lambda_r| \geq \varepsilon_\lambda > |\lambda_{r+1}|$ for some small tolerance $\varepsilon_\lambda < 1$. Then the Hessian-based subspace spanned by the eigenvectors $\Psi = (\psi_1, \ldots, \psi_r)$ captures the most variation of the parameter $x$ informed by data $y$. We remark that to solve the generalized Hermitian eigenvalue problem (29), we employ a randomized SVD algorithm [17], which requires $O(NrC_h + dr^2)$ flops, where $C_h$ is the cost of a Hessian action in a direction.

### 3.4 Parallel and adaptive pSVN algorithm

Given the bases $\Psi$ as the data-informed parameter directions, we can draw samples $x_1, \ldots, x_N$ from the prior distribution and push them by pSVN to match the posterior distribution in a low-dimensional subspace, while keeping the components of the samples in the complementary subspace unchanged. We set the stopping criterion as: (i) the maximum norm of the updates $w_m^l - w_m^{l-1}$, $m = 1, \ldots, N$, is smaller than a given tolerance $\text{Tol}_g$; (ii) the maximum norm of the gradients $g_m$, $m = 1, \ldots, N$, is smaller than a given tolerance $\text{Tol}_w$; or (iii) the number of iterations $l$ reaches a preset number $L$. Moreover, we take advantage of pSVN advantages in low-dimensional subspaces—including fast computation, lightweight communication, and low memory footprint—and provide an efficient parallel implementation using MPI communication in Algorithm 1, with analysis in Appendix C.

---

**Algorithm 1** pSVN in parallel using MPI

---

1: **Input:** $M$ prior samples, $x_1, \ldots, x_M$, in each of $K$ cores, bases $\Psi$, and density $p_y$ in all cores.
2: **Output:** posterior samples $x_1^y, \ldots, x_M^y$ in each core.
3: Perform projection (13) to get $x_m = x_m^r + x_m^\perp$ and the samples $w_m^{l-1}$, $m = 1, \ldots, M$, at $l = 1$.
4: *Perform* MPI_Allgather *for* $w_m^{l-1}$, $m = 1, \ldots, M$.
5: **repeat**
6:     Compute the gradient and Hessian by (25).
7:     *Perform* MPI_Allgather *for the gradient and Hessian.*
8:     Compute the kernel and its gradient by (12) and (26).
9:     *Perform* MPI_Allgather *for* $k_m$, $m = 1, \ldots, M$,
    MPI_Allreduce *w. sum for* $\sum_m k_m$ *and* $\sum_m \nabla_w k_m$.
10:     Assemble and solve system (27) for $c_1, \ldots, c_M$.
11:     Perform a line search to get $w_1^l, \ldots, w_M^l$.
12:     *Perform* MPI_Allgather *for* $w_m^l$, $m = 1, \ldots, M$.
13:     Update the samples $x_m^r = \Psi w_m^l + \bar{x}$, $m = 1, \ldots, M$.
14:     Set $l \leftarrow l + 1$.
15: **until** A stopping criterion is met.
16: Reconstruct samples $x_m^y = x_m^r + x_m^\perp$, $m = 1, \ldots, M$.

---

In Algorithm 1, we assume that the bases $\Psi$ for the projection are the data informed parameter directions, which are obtained by the Hessian-based algorithm in Section 3.3 at the "representative" samples $x_1, \ldots, x_N$. However, we do not have these samples but only the prior samples at the beginning. To address this problem, we propose an adaptive algorithm that adaptively construct the bases $\Psi$ based on samples pushed forward from the prior to the posterior, see Algorithm 2.

---

**Algorithm 2** Adaptive pSVN

---

1: **Input:** $M$ prior samples, $x_1, \ldots, x_M$, in each of $K$ cores, and density $p_y$ in all cores.
2: **Output:** posterior samples $x_1^y, \ldots, x_M^y$ in each core.
3: Set level $l_2 = 1$, $x_m^{l_2-1} = x_m$, $m = 1, \ldots, M$.
4: **repeat**
5:     Perform the eigendecomposition (29) at samples $x_1^{l_2-1}, \ldots, x_M^{l_2-1}$, and form the bases $\Psi^{l_2}$.
6:     Apply **Algorithm** 1 to update the samples
    $[x_1^{l_2}, \ldots, x_M^{l_2}] = \text{pSVN}([x_1^{l_2-1}, \ldots, x_M^{l_2-1}], K, \Psi^{l_2}, p_y)$.
7:     Set $l_2 \leftarrow l_2 + 1$.
8: **until** A stopping criterion is met.

---

## 4 Numerical experiments

We demonstrate the convergence, accuracy, and dimension-independence of the pSVN method by two examples, one a linear problem with Gaussian posterior to demonstrate the convergence and accuracy of pSVN in comparison with SVN and SVGD, the other a nonlinear problem to demonstrate accuracy as well as the dimension-independent and sample-independent convergence of pSVN and its scalability w.r.t. the number of processor cores. The code is described in Appendix D.

## 4.1 A linear inference problem

For the linear inference problem, we have the parameter-to-observable map

$$f(x) = Ax, \tag{31}$$

where the linear map $A = O(Bx)$, with an observation map $O : \mathbb{R}^d \to \mathbb{R}^s$, and an inverse discrete differential operator $B = (L + M)^{-1} : \mathbb{R}^d \to \mathbb{R}^d$ where $L$ and $M$ are the discrete Laplacian and mass matrices in the PDE model $-\triangle u + u = x$, in $(0, 1)$, $u(0) = 0$, $u(1) = 1$. $s = 15$ pointwise observations of $u$ with $1\%$ noise are distributed with equal distance in $(0, 1)$. The input $x$ is a random field with Gaussian prior $\mathcal{N}(0, \Gamma_0)$, where $\Gamma_0$ is discretized from $(I - 0.1\triangle)^{-1}$ with identity $I$ and Laplace operator $\triangle$. We discretize this forward model by a finite element method with piecewise linear elements on a uniform mesh of size $2^n$, which leads to the parameter dimension $d = 2^n + 1$.

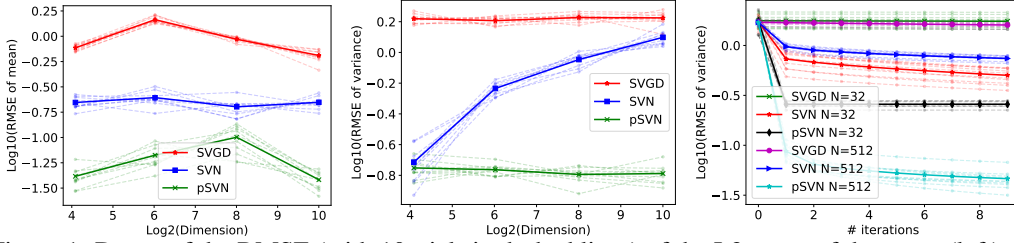

Figure 1: Decay of the RMSE (with 10 trials in dashed lines) of the L2-norm of the mean (left) and pointwise variance (middle) of the parameter w.r.t. dimension $d = 16, 64, 256, 1024$ with $N = 128$ samples. Right: Decay of the RMSE of the L2-norm of the pointwise variance with $N = 32, 512$ samples in parameter dimension $d = 256$ w.r.t. # iterations. Comparison for SVGD, SVN, pSVN.

Figure 1 compares the convergence and accuracy of SVGD, SVN, and pSVN by the decay of the root mean square errors (RMSE) (using 10 trials and 10 iterations) of the sample mean and variance (with L2-norm of errors computed against analytic values in (30)) w.r.t. parameter dimensions and iterations. We observe much faster convergence and greater accuracy of pSVN relative to SVGD and SVN, for both mean and especially variance, which measures the goodness of samples. In particular, we see from the middle figure that the SVN estimate of variance deteriorates quickly with increasing dimension, while pSVN leads to equally good variance estimate. Moreover, from the right figure we can see that pSVN converges very rapidly in a subspace of dimension 6 (at tolerance $\varepsilon_\lambda = 0.01$ in Section 3.3, i.e., $|\lambda_7| < 0.01$) and achieves higher accuracy with larger number of samples, while SVN converges slowly and leads to large errors. With the same number of iterations of SVN and pSVN, SVGD produces no evident error decay.

## 4.2 A nonlinear inference problem

We consider a nonlinear benchmark inference problem (which is often used for testing high-dimensional inference methods [30, 12, 3]), whose forward map is given by $f(x) = O(S(x))$, with observation map $O : \mathbb{R}^d \to \mathbb{R}^s$ and a nonlinear solution map $u = S(x) \in \mathbb{R}^d$ of the lognormal diffusion model $-\nabla \cdot (e^x \nabla u) = 0$, in $(0, 1)^2$ with $u = 1$ on top and $u = 0$ on bottom boundaries, and zero Neumann conditions on left and right boundaries. 49 pointwise observations of $u$ are equally distributed in $(0, 1)^2$. We use $10\%$ noise to test accuracy against a DILI MCMC method [12] with 10,000 MCMC samples as reference and a challenging $1\%$ noise for a dimension-independence test of pSVN. The input $x$ is a random field with Gaussian prior $\mathcal{N}(0, \Gamma_0)$, where $\Gamma_0$ is a discretization of $(I - 0.1\triangle)^{-2}$. We solve this forward model by a finite element method with piecewise linear elements on a uniform mesh of varying sizes, which leads to a sequence of parameter dimensions.

Figure 2 shows the comparison of the accuracy and convergence of pSVN and SVN for their sample estimate of mean and variance. We can see that in high dimension, $d = 1089$, pSVN converges faster and achieves higher accuracy than SVN for both mean and variance estimate. Moreover, SVN using the kernel (12) in high dimensions (involving low-rank decomposition of the metric $M$ for high-dimensional nonlinear problems) is more expensive than pSVN per iteration.

We next demonstrate pSVN's independence of the number of parameter and sample dimensions, and its scalability w.r.t. processor cores. First, the dimension of the Hessian-based subspace $r$, which determines the computational cost of pSVN, depends on the decay of the absolute eigenvalues $|\lambda_i|$

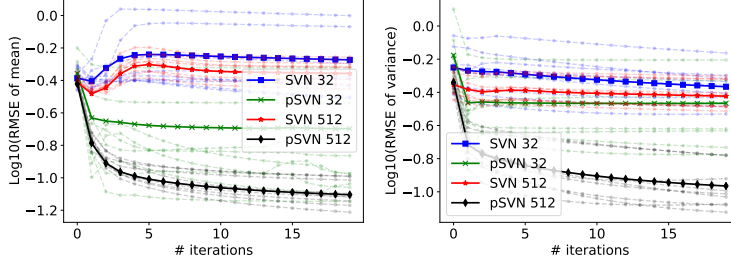

Figure 2: Decay of the RMSE (with 10 trials in dashed lines) of the L2-norm of the mean (left) and pointwise variance (right) of the parameter with dimension $d = 1089$ and $N = 32, 512$ samples.

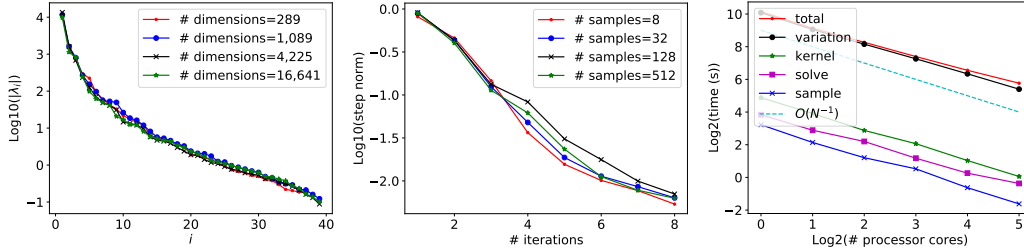

Figure 3: Left: Decay of eigenvalues $\log_{10}(|\lambda_i|)$ with increasing dimension $d$. Middle: Decay of a stopping criterion—the averaged norm of the update $w^l - w^{l-1}$ w.r.t. the iteration number $l$, with increasing number of samples. Right: Decay of the wall clock time (seconds) of different computational components w.r.t. increasing number of processor cores on a log-log scale.

as presented in Section 3.3. The left part of Figure 3 shows that with increasing $d$ from 289 to over 16K, $r$ does not change, which implies that the convergence of pSVN is independent of the number of nominal parameter dimensions. Second, as shown in the middle part of Figure 3, with increasing number of samples for a fixed parameter dimension $d = 1089$, the averaged norm of the update $w^l - w^{l-1}$, as one convergence indicator presented in Subsection 3.4, decays similarly, which demonstrates the independence of the convergence of pSVN w.r.t. the number of samples. Third, in the right of Figure 3 we plot the *total* wall clock time of pSVN and the time for its computational components in Algorithm 1 using different number of processor cores for the same work, i.e., the same number of samples (256), including *variation* for forward model solve, gradient and Hessian evaluation, as well as eigendecomposition, *kernel* for kernel and its gradient evaluation, *solve* for solving the Newton system (27), and *sample* for sample projection and reconstruction. We can observe nearly perfect strong scaling w.r.t. increasing number of processor cores. Moreover, the time for *variation*, which depends on parameter dimension $d$, dominates the time for all other components, in particular *kernel* and *solve* whose cost only depends on $r$, not $d$.

## 5   Conclusion

We presented a fast and scalable variational method, pSVN, for Bayesian inference in high dimensions. The method exploits the geometric structure of the posterior via its Hessian, and the intrinsic low-dimensionality of the change from prior to posterior characteristic of many high-dimensional inference problems via low rank approximation of the averaged Hessian of the log likelihood, computed efficiently using randomized matrix-free SVD. The fast convergence and higher accuracy of pSVN relative to SVGD and SVN, its complexity that is independent of parameter and sample dimensions, and its scalability w.r.t. processor cores were demonstrated for linear and nonlinear inference problems. Investigation of pSVN to tackle intrinsically high-dimensional inference problem (e.g., performed in local dimensions as the message passing scheme or combined with dimension-independent MCMC to update samples in complement subspace) is ongoing. Further development and application of pSVN to more general probability distributions, projection basis constructions, and forward models such as deep neural network, and further analysis of the convergence and scalability of pSVN w.r.t. the number of samples, parameter dimension reduction, and data volume, are of great interest.

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
