[Supplementary Material]

## Appendix A: Proof of Theorem 1

*Proof.* By definition of the posterior density $p(x|y)$ in (2) and the projected posterior density $p^r(x|y)$ in (14), we have

$$
\begin{aligned}
\mathcal{D}_{\mathrm{KL}}(p(x|y)\,|\,p^r(x|y)) &= \int_{\mathbb{R}^d} \log\left(\frac{p_y(x)}{p_y^r(x)}\frac{Z_r}{Z}\right)\frac{1}{Z}p_y(x)dx \\
&= \int_{\mathbb{R}^d}(\eta_y(x^r) - \eta_y(x))\frac{1}{Z}p_y(x)dx + \log\left(\frac{Z_r}{Z}\right),
\end{aligned}
\tag{32}
$$

where we used the definitions of $p_y(x)$ and $p_y^r(x)$ in (2) and (14) in the second equality. By definition of $\eta_y$ in (3), we have

$$
\begin{aligned}
\eta_y(x^r) - \eta_y(x) &= \frac{1}{2}||y - f(x^r)||_\Gamma^2 - \frac{1}{2}||y - f(x)||_\Gamma^2 \\
&= y^T\Gamma^{-1}(f(x) - f(x^r)) - \frac{1}{2}(f(x) + f(x^r))^T\Gamma^{-1}(f(x) - f(x^r)) \\
&\le ||y^T||_\Gamma||f(x) - f(x^r)||_\Gamma + \frac{1}{2}||f(x) + f(x^r)||_\Gamma||f(x) - f(x^r)||_\Gamma \\
&\le \frac{C_b}{2}(2||y^T||_\Gamma + ||f(x)||_\Gamma + ||f(x^r)||_\Gamma)||x - x^r||_X
\end{aligned}
\tag{33}
$$

where we used Assumption 1 in the second inequality for $\max\{||x||_X, ||x^r||_X\} < b$. Therefore, the first integral in (32), denoted as (I) can be bounded by (note that $\exp(-\eta_y(\cdot)) \le 1$)

$$
(I) \le \frac{C_b}{2Z}\int_{\mathbb{R}^d}\left(2||y^T||_\Gamma + ||f(x)||_\Gamma + ||f(x^r)||_\Gamma\right)p_0(x)dx\,||x - x^r||_X,
\tag{34}
$$

By Assumption 1, we have

$$
(I) \le C_I||x - x^r||_X,
\tag{35}
$$

for a constant $C_I = C_b(||y^T||_\Gamma + C_f)/Z$.

For the second term $\log(Z_r/Z)$ in (32), we have for

$$
\begin{aligned}
\left|1 - \frac{Z_r}{Z}\right| &= \frac{1}{Z}|Z - Z_r| \\
&\le \frac{1}{Z}\int_{\mathbb{R}^d}|\exp(-\eta_y) - \exp(-\eta_y^r)|p_0(x)dx \\
&\le \frac{1}{Z}\int_{\mathbb{R}^d}|\eta_y - \eta_y^r|p_0(x)dx \\
&\le C_I||x - x^r||_X,
\end{aligned}
\tag{36}
$$

where in the second inequality we used that $|e^{-\tau_1} - e^{-\tau_2}| < |\tau_1 - \tau_2|$ for $\tau_1, \tau_2 > 0$, for the last inequality we used the bound of the first integral of (32). Then by $\log(1 + \tau) \le \tau$ for $\tau \ge 0$, we have

$$
\log\left(\frac{Z_r}{Z}\right) \le \log\left(1 + \left|\frac{Z_r}{Z} - 1\right|\right) \le \left|\frac{Z_r}{Z} - 1\right| \le C_I||x - x^r||_X,
\tag{37}
$$

which completes the proof with constant $C = 2C_I$

$\square$

## Appendix B: Globalization by line search

Except for in the case of a linear inference problem, the cost functional—Kullback–Leibler divergence—is nonconvex. In the case of that the Newton approximation to the Kullback–Leibler divergence is locally exact, the simple choice of $\varepsilon = 1$ is the optimal choice for the step size. However, since the geometry generally exhibits complex non-quadratic local structure, a constant stepsize $\varepsilon$ renders minimization of $\mathcal{D}_{\mathrm{KL}}$ inefficient. A careful choice of the step size $\varepsilon$ is crucial for both fast convergence and stability of Stein variational methods. While, there are many options to choose from,

we employ an Armijo line search globalization method to choose this step size, to much success. Specifically, at step $l = 1, 2, \ldots$, we seek $\varepsilon$ to minimize the Kullback–Leibler divergence

$$\mathcal{D}_{\mathrm{KL}}((T_l)_* \pi_{l-1} | \pi_y) = \mathcal{D}_{\mathrm{KL}}(\pi_{l-1} | (T_l)^* \pi_y), \tag{38}$$

where $(T_l)^*$ is the pullback operator. Because

$$\begin{aligned}
\mathcal{D}_{\mathrm{KL}}(\pi_{l-1} | (T_l)^* \pi_y) &= \mathbb{E}_{\pi_{l-1}}[\log(\pi_{l-1}(\cdot))] \\
&- \mathbb{E}_{\pi_{l-1}}[\log(\pi_y(T_l(\cdot)) | \det \nabla_w T_l(\cdot) |)],
\end{aligned} \tag{39}$$

where the first term does not depend on $\varepsilon$. Hence we only need to consider the second term denoted as $\mathcal{D}_{\mathrm{KL}}^{(2)}$, which is evaluated by the sample average approximation as

$$\begin{aligned}
\mathcal{D}_{\mathrm{KL}}^{(2)} \approx &- \frac{1}{N} \sum_{n=1}^{N} \log(\pi_y(T_l(w_n^{l-1}))) \\
&- \frac{1}{N} \sum_{n=1}^{N} \log(| \det \nabla_w T_l(w_n^{l-1}) |),
\end{aligned} \tag{40}$$

which can be readily computed for every $\varepsilon$. We remark that the second term of (40) is close to 0 when the kernel function $k_n(w)$ in (21) is close to 0 at every sample $w_m^{l-1}$ for $m \neq n$, so we only need to consider the first term of (40). Moreover, to guarantee that $\mathcal{D}_{\mathrm{KL}}^{(2)}$ is reduced for a suitable $\varepsilon$, we can find sample-dependent step sizes $\varepsilon(w_n^{l-1})$ such that

$$- \log(\pi_y(T_l(w_n^{l-1}))) \tag{41}$$

is reduced for each $n = 1, \ldots, N$.

## Appendix C: Complexity analysis for parallel pSVN

We presented a parallel implementation of pSVN in Algorithm 1. Lines 4 and 12 involve global communication(gather and broadcast) of the low-dimensional samples $w_m$, $m = 1, \ldots, M$, of size $Mr$, which are used for the kernel and its gradient evaluations at all samples, as well as for the sample update in (21). Line 7 involves global communication (gathers and broadcasts) of the gradients (of size $Mr$) and Hessians (of size $Mr^2$) of the log posterior density (25), which are used in the expectation evaluation at all samples for assembling the system (27). Line 9 involves global communication (gathers and broadcasts) of the kernel values (of size $NM$) at all samples, which are used in moving the samples by (21). Meanwhile, Line 9 gathers a local sum of the kernel values $\sum_m k_m(w)$ (of size $N$) and its gradients $\sum_m \nabla_w k_m(w)$ (of size $rN$), performs a global sum of them, and broadcasts the results to all cores, which are used for assembling the lumped Hessian (28). In summary, the data volumes of communication in Algorithm 1 are bounded by $\max(Mr^2, MN)$ floats.

To implement a parallel version of the adaptive pSVN Algorithm 2, we only need to construct the bases $\Psi$ in parallel to replace its Line 5, for which we perform an averaged Hessian action in random directions with $M$ samples in each core by $O(M(rC_h))$ flops, followed by a MPI_Allreduce with a SUM operator to get a global averaged Hessian action before performing randomized SVD with $O(dr^2)$ flops. The data volumes for communication is $dr$ floats, which dominates all other communication cost if $d$ is so large that $dr > \max(r^2M, NM)$. Alternatively, we can construct the bases $\Psi$ using Hessian at the local samples in each core without communication for $\Psi$.

## Appendix D: Bayesian Autoencoder Example

We consider a Bayesian inference problem constrained by a convolutional autoencoder neural network.

In the Bayesian autoencoder problem, we seek to learn a low dimensional representation of data under uncertainty. Given input data $z \in \mathbb{R}^{\mathrm{data}}$ the $2m$ layer autoencoder mapping is defined as

$$y(\cdot) = o_{i=1}^{2m} \phi_i(w_i * (\cdot) + b_i) \tag{42}$$

where $w_i$ is the convolution kernel (weights) for layer $i$, and $\phi_i$ is an nonlinear activation functions. The $*$ operations represents both convolution and downsampling. The first $m$ compositions map

down to a low dimensional latent representation of the input data $z$, the last $m$ compositions map the data back to $\mathbb{R}^{\text{data}}$.

The data $z$ for the problem are 1000 randomly selected MNIST images. The target data has 5% i.i.d. noise added to it based on min-max normalization of the data. The objective function for the autoencoder training problem is a least squares misfit that measures the error between the reconstructed input image and the noisy target image. The inference parameter $\{x_i\} = \{(w_i, b_i)\} \in \mathbb{R}^d$ has the i.i.d. prior $\mathcal{N}(0, \sigma_i^2)$, where $\sigma_1 = 1$, and $\sigma_{i+1}^2 = 0.5\sigma_i^2$. We use a fixed convolution kernel support of $4 \times 4$ and vary the number of filters on each layer from $2, 4, 8$ and use $m = 2$ layers.

Low rank structure of Hessians has been observed for neural network training problems [1, 15, 25]. Due to the low dimensional nature of the autoencoder, the pSVN algorithm can efficiently find a $r$ dimensional Hessian subspace. The dimensionality of this subspace depends on the decay of the absolute eigenvalues $|\lambda_i|$.

Numerical results are shown below in Figure 4. In these trials the problem dimension of the inference parameter is 133; 128 particles were used. A fixed candidate rank was chosed to be $r = 40$, which is the effective rank of the prior preconditioned Hessian for the problem as seen in the left figure in 4. The right figure shows that pSVN minimizes the objective function in training faster than the SVN algorithm for this particular example.

Figure 4: Left: Absolute value of eigenvalues of the prior preconditioned Hessian used for the pSVN subspace. Right: Training error for pSVN vs SVN.

## Appendix E: Code

We implemented the stein variational methods (and the DILI MCMC method) in hIPPYlib (`https://hippylib.github.io/`), a python library for solving inverse problems, which relies on FEniCS (`https://fenicsproject.org/`), a computing platform for solving partial differential equations. The code for our tests can be downloaded from `https://github.com/cpempire/pSVN`.