[Reviews · NeurIPS 2019]

Reviewer 1



Convergence of existing Stein variational methods is known to suffer in high dimensions due to the locality of the kernel. The authors address this problem by exploiting the structure of the posterior distribution. Concretely, they propose to perform Stein gradient steps in a low-dimensional projection subspace. The basis of the projection space is derived from the expected Hessian of the log-likelihood, where the expectation is adaptively approximated by an empirical estimate. The introduced projection scheme and the corresponding Stein gradient steps are well motivated and presented. A theoretical analysis is presented to bound the bias introduced by the projection. Empirical experiments are performed to validate the effectiveness of the method for a linear and a non-linear inference problem. Some remarks: - The paper mentions a few important details without further discussing/studying them. For example, the authors assume that the update from the prior to the posterior in the subspace complementary to the projection subspace is negligible. However, there is no discussion or experiments whether/when this assumption holds, and what's the impact on the introduced bias (i.e., Theorem 1). What if the approximation of the avg. Hessian is poor, e.g., because a (too) uninformative prior? Similarly, Algorithm 2 (Adaptive pSVN) is a bootstrapping version of pSVN which is presented with a motivating idea only; no further analysis or experiments. - The presentation and theoretical analysis is restricted to Gaussian likelihoods, whereas the proposed method should be applicable to a broader class of densities (essentially all inference tasks with differentiable log-densities). The method could be presented for this more general case (keeping the analysis for the Gaussian likelihood). - The notation and text does not clearly distinguish mappings and arguments; e.g., the Fr├ęchet derivate and the preconditioner in Eq. 7 are mappings from R^d into R^d and R^(dxd), respectively; gradient g (line 81) and projection x^r (Eq. 13) are mappings, too. Similarly, mappings and operators are mixed; cf. Eq 5 and 6. It should also be noted that H_{mn} in Eq. 11 denotes a dxd dimensional matrix rather than a single entry of matrix H. While for most parts of the text, the meaning is clear from the context, I would prefer a more stringent notation. - The experiments focus on a simple linear Gaussian inference problem, and a non-linear inverse problem. It would be interesting to evaluate the method for other models such as hierarchical and mixture models. The method may also work for non-Gaussian likelihoods.

Reviewer 2



The author(s) propose a variational method, pSVN. Compared to its parent algorithms, SVGD and SVN, pSVN is faster to converge, has higher accuracy, and offers a complexity independent of parameter and sample dimensions. The authors do a very nice job of covering the basics of Stein variational methods before diving into the derivation and exposition of their algorithm. Overall, the theory is well presented and seemingly sound. I believe there is a small typo (extra 'is') in lines 98-99. The experiments are well explained. Their results support the claimed advantages of accuracy, speed, and scalability that pSVN has over SVGD and SVN. I was impressed that the author(s) also experimentally characterized the speed gains of the parallelization available to their algorithm. My only complaint with this paper, is the examples, particularly the nonlinear one, seem contrived and perhaps overly suited to their proposed methdod. The first example being linear makes sense since the authors point out that Hessian-based subspace is optimal for linear f (line 173). Although, from their conclusion it seems these kinds of broader applications are being actively worked on. Overall, I am quite impressed with the paper, but am left wondering about broader applicability given the limited experimental constructions. Edit: I've read the response and other reviews and look forward to this being accepted :)

Reviewer 3



The author proposed a projected Stein variational Newton (pSVN) method for high-dimensional Bayesian inference. The author employed Hessian of the log posterior to explore the low dimension geometric structure of posterior distribution to address the curse of dimensionality. Experiment results on both linear and nonlinear synthetic data are presented. Overall, the paper is well organized and the presentation is clear. Detail comments and suggestions are as following. - Using Hessian to find the low dimensional projection direction is under Gaussian assumption. Will this Gaussian assumption limit the ability of proposed method? - In the nonlinear inference problem section, the dimension of the Hessian-based subspace r does not change with the increase of sample dimension. Another possibility is the data generation model is too simple. One experiment using more complex generative model will be able to rule out this possibility? - For these two example, it may not be a bad idea to have MCMC methods as one of the baseline (or ground truth) Minor comment: I really like the right figure in figure 3. This show the potential of using proposed method in real world large scale problem. I have read author's rebuttal and I will keep my original scores.

[Author Response · NeurIPS 2019]

We are grateful to all the reviewers for their careful and overall positive assessment of our manuscript, and in particular
thank for their helpful suggestions for its improvement. As the reviewers pointed out, convergence of existing Stein
variational methods is known to suffer in high dimensions. To address this critical challenge, we proposed the algorithm
pSVN by exploiting the intrinsic low-dimensionality of the difference between the prior and posterior, which can be
observed in many applications and proved in some cases as in [1, 4, 5, 6, 8, 9, 11, 16, 26] and references therein. As
the reviewers assessed, the algorithm is well motivated and presented with concrete theoretical analysis and empirical
validation, which is shown to converge faster and achieve higher accuracy compared to SVGD and SVN for both the
tested linear and nonlinear problems, as well as to offer a complexity independent of parameters and sample dimensions,
with the parallel scalability demonstrated to pose the potential of being used in real world large scale problem. We
really appreciate the reviewers' careful reading, deep understanding, and high recognition of pSVN's merits.

Below are our responses for the helpful questions and suggestions of each reviewer.

**To Reviewer 1:** (1) For Gaussian priors, $x^\perp$ is in fact independent of $x^r$, i.e., $p_0(x) = p_0^r(x^r)p_0^\perp(x^\perp)$, so there is no
need to update $x^\perp$ (no bias in Theorem 1 by freezing $x^\perp$). For non-Gaussian priors, in general $x^\perp$ does depend on $x^r$,
even negligible for the posterior update if the dimension of the projected subspace $r$ is sufficiently large. The adaptive
pSVN with adaptively changing subspaces can in fact enable data-informed update in $x^\perp$ at different steps, i.e., the
freezing is only effective in the same projection space. We will add these clarifications in the revision. More subtle
convergence analysis for adaptive pSVN will be studied in future work. The adaptive pSVN was indeed used in the
nonlinear test problem. We will add an empirical comparison with pSVN in the revision. The approximation of the
averaged Hessian is also performed adaptively depending on the current particles, which becomes less dependent on the
prior (even uninformative) when the particles approach the posterior. (2) Indeed, the method works for more general
cases than Gaussian likelihoods as long as the posterior density is differentiable. We will update the presentation of the
method to the more general cases and keep the theoretical analysis to the Gaussian likelihoods in this work. (3) Thank
you for pointing out these (slightly) abused/unclear notations. We will revise them accordingly. (4) We add a new
experiment on Bayesian autoencoder networks to demonstrate the crucial property of the intrinsic low-dimensionality
exploited by pSVN, see below, which will be added to the supplementary material with more details in the revision.

**To Reviewer 2:** We appreciate very much the reviewer's positive and detailed assessment of our work. The algorithm
works admittedly particularly well for the two examples using Gaussian priors with compact covariances, which are
commonly found in many application areas, e.g., spatial statistics, geostatistis, physical cosmology, etc. [Lindgren and
Rue, 2011]. To test on more general models, we add an experiment on Bayesian autoencoder networks to demonstrate
the crucial property of the intrinsic low-dimensionality exploited by pSVN, see below. We will add more details of this
experiement in the revision.

**To Reviewer 3:** (1) We understand that the current presentation and examples may be a bit misleading that using
Hessian to find the low-dimensional projection direction is under Gaussian assumption, which is however not true. The
covariance $\Gamma_0$ in (30) does not require a Gaussian prior. Moreover, for general priors, e.g., uniform, we can also ignore
the covariance $\Gamma_0$ in (30), as long as we can compute the Hessian of the (log) posterior density. We clarify this in the
revision. (2) We agree that the dimension of the Hessian-based subspace $r$ does not change with increasing parameter
dimension makes the generation model looks too simple, which is however often true in applications when the Gaussian
prior covariance is compact. We add an experiment of Bayesian autoencoder networks with different eigenvalue decays
to rule out this impression, see below. (3) The suggestion to use MCMC as ground truth is valuable. In fact, we did use
MCMC, page 7, line 227, in the nonlinear problem as the baseline to test the accuracy of the algorithm. For the linear
problem, the posterior is explicitly given as in (31), which was used as the baseline instead of MCMC samples.

We briefly present an experiment on Bayesian autoencoder neural network model to demonstrate the intrinsic low-dimensionality exploited by pSVN. The four cases with increasing parameter dimension (775, 3,079, 5,395, 21,523) correspond to autoencoders of convolutional neural networks with (4, 4, 6, 6) layers, each layer with (6, 6, 6, 6) convolutional kernels of dimension (8, 16, 8, 16), respectively. To compute the eigenvalues of the Hessian of the log-likelihood, we use 10,000 MNIST images as the data corrupted with i.i.d. additive noise of 5% noise-to-signal ratio, and i.i.d. Gaussian prior $\mathcal{N}(0, \sigma_i^2)$ with the first layer variance set to 1 and subsequent layers decay by a constant 0.5 multiplicative factor. The right figure displays the dominate eigenvalues $|\lambda_i|$, which converge rapidly with over 1000X reduction for the first 100 eigenvalues (sharp decay in last few due to artifact of randomized SVD using 100 samples), which indicates the intrinsic low-dimensional structure. We will add more details of pSVN for this experiment as the supplementary material in the revision.

Decay of eigenvalues for autoencoders with increasing parameter dimension

[Meta-Review · NeurIPS 2019]

The reviewers agree that this submission represents an important contribution to the field. Please be sure to carefully review and address the concerns of all reviewers in the revision.